

# Leveraging a hybrid convolutional gated recursive diabetes prediction and severity grading model through a mobile app

Alhuseen Omar Alsayed[1], Nor Azman Ismail[1], Layla Hasan[1], Muhammad Binsawad[2] and Farhat Embarak[3]

[1] Faculty of Computing, Universiti Teknologi Malaysia, Johor Bahur, Johor, Malaysia
[2] Department of Information Systems, Faculty of Computing and Information Technology, King Abdul Aziz University, Jeddah, Makkah, Saudi Arabia
[3] Faculty of Information and Technology, University of Ajdabiya, Ajdabiya, Libya

Corresponding author
Alhuseen Omar Alsayed,
nuriy3@graduate.utm.my

## ABSTRACT

Diabetes mellitus is a common illness associated with high morbidity and mortality rates. Early detection of diabetes is essential to prevent long-term health complications. The existing machine learning model struggles with accuracy and reliability issues, as well as data imbalance, hindering the creation of a dependable diabetes prediction model. The research addresses the issue using a novel deep learning mechanism called convolutional gated recurrent unit (CGRU), which could accurately detect diabetic disorder and their severity level. To overcome these obstacles, this study presents a brand-new deep learning technique, the CGRU, which enhances prediction accuracy by extracting temporal and spatial characteristics from the data. The proposed mechanism extracts both the spatial and temporal attributes from the input data to enable efficient classification. The proposed framework consists of three primary phases: data preparation, model training, and evaluation. Specifically, the proposed technique is applied to the BRFSS dataset for diabetes prediction. The collected data undergoes pre-processing steps, including missing data imputation, irrelevant feature removal, and normalization, to make it suitable for further processing. Furthermore, the pre-processed data is fed to the CGRU model, which is trained to identify intricate patterns indicating the stages of diabetes. To group the patients based on their characteristics and identity patterns, the research uses the clustering algorithm which helps them to classify the severity level. The efficacy of the proposed CGRU framework is demonstrated by validating the experimental findings against existing state-of-the-art approaches. When compared to existing approaches, such as Attention-based CNN and Ensemble ML model, the proposed model outperforms conventional machine learning techniques, demonstrating the efficacy of the CGRU architecture for diabetes prediction with a high accuracy rate o f 99.9%. Clustering algorithms are more beneficial as they help in identifying the subtle pattern in the dataset. When compared to other methods, it can lead to more accurate and reliable prediction. The study highlights how the cutting-edge CGRU model enhances the early detection and diagnosis of diabetes, which will eventually lead to improved healthcare outcomes. However, the study limits to work on diverse datasets, which is the only thing considered to be the drawback of this research.

# INTRODUCTION

Currently, diabetes mellitus (DM) has developed into a very serious condition. Numerous people have DM, which is categorized as a "non-communicable disease (NCB)." According to 2017 statistics, an estimated 425 million people worldwide are predicted to have diabetes, a condition that kills between two and five million people annually. By 2045, it is anticipated to reach 629 million (*Kalyankar, Poojara & Dharwadkar, 2017*). Insulin-dependent type-1 diabetes that falls within the category of diabetes mellitus (DM) is also known as diabetes mellitus (IDDM). Patients with this type of diabetes require insulin injections because their bodies cannot produce enough insulin. Type 2 diabetes, on the other hand, is known as non-insulin-dependent diabetes (NIDDM). When cells in the body are unable to utilize insulin as intended, they develop this kind of diabetes. Gestational diabetes, or type-3 diabetes, is commonly diagnosed in pregnant women who did not have diabetes before pregnancy. Additionally, a diabetic has a higher risk of developing various health issues (*Mujumdar & Vaidehi, 2019*). Since diabetes is a major global health concern with a high morbidity and mortality rate, there is a need for early detection of diabetes. Hence, this research is motivated to provide an effective mechanism for diabetes detection and accurate severity level prediction for timely intervention and prevention of long-term complications.

Recent scientific and engineering advancements in technology have driven various applications of artificial intelligence (AI), such as voice recognition, self-driving cars, pattern matching, recommendation systems, real estate and stock market predictions, and healthcare. The field of biocomputing has several uses for the use of AI, including analytics, cancer categorization, diabetic kidney condition evaluation, and the prediction of cardiac attacks (*Alhuseen et al., 2023*). Through machine learning, the system gains improved performance by learning from its past experiences. AI systems are applied in handwriting identification, speech detection, facial recognition, responding to consumer queries, as well as control, organization, and planning tasks. Nowadays, the healthcare industry relies heavily on machine learning to anticipate a range of illnesses (*Alsayed et al., 2023*). Thus, using machine learning algorithms for the early diagnosis of diabetes is crucial for improving human life expectancy (*Alsayed et al., 2021*). For instance, *Vhaduri & Prioleau (2020)* revealed that using personal health devices for continuous glucose monitoring in diabetes care can facilitate early detection.

Machine learning is regarded as a key technique in diabetes care because it can analyze large datasets and identify intricate patterns that are difficult to detect manually. To accurately predict the development of diabetes and identify high-risk individuals, researches can employ machine learning models, which also facilitate early intervention and improved health outcomes. Machine learning can process vast amounts of data, making it more efficient and ideal for the healthcare system (*Alsayed, Ismail & Hasan, 2024*). Thus, it can be stated that predictions done by machine learning models can provide

accurate and reliable outcomes and they can also identify the patterns or characteristics that are missed by the professionals. Moreover, the machine learning model is visible to additional data, allowing it to adjust and provide more accurate results over time. This can increase their efficacy in identifying and diagnosing diabetes. The application of machine learning models in the prediction or diagnosis of diabetes can help control and treat the condition at an early stage, which is considered a revolution in machine learning (*Modi, Kumar & Geetha, 2023*).

The creation and implementation of an artificial intelligence software system that considers diabetes management is described in this work (*Alotaibi et al., 2014*). The creation of a meal suggestion structure, activity tracking and monitoring, chatbots for diabetes education, and medications reminders are key priorities in diabetes management. A regression approach is used in the prediction system presented in a study by *Orabi, Kamal & Rabah (2016)* to determine the age at which an individual is likely to develop diabetes based on 23 factors (*Rasool et al., 2022a*, *2022b*). *Alghamdi et al. (2017)* presented a study on the effectiveness of various machine-learning techniques for predicting hyperglycemia from medical data. The proposed work was driven primarily by the need for a program that anyone could use to estimate the likelihood of developing diabetes.

Over the past decade, artificial intelligence and machine learning have become popular in many disciplines, especially in the medical area (*Bajwa et al., 2021*). Although there is evidence showing how AI and ML assist in diagnosing diseases including diabetes, much of the research focuses on the positive impact of these technologies, while giving less attention to the existing drawbacks in current models. While AI-based methods can be highly accurate in predicting diabetes, challenges remain in predicting severity levels, which are crucial for determining the appropriate treatment program for patients (*Ramudu et al., 2023*). Many current approaches experience problems in processing large datasets when temporal and spatial data analysis is needed with various problems related to real-time applicability, scalability, or the reliability of predictions (*Wang et al., 2024*). In addition, the translation of these superior models into easily accessible, mobile-responsive tools and applications has received limited attention, reducing the availability of effective alternatives for patients and clinicians (*Haleem et al., 2021*). Existing studies on diabetes prediction using machine learning have struggled with accuracy and reliability due to the disease's complexity. Inspired by the deep learning algorithm and aiming to address gaps in existing research, this study develops a hybrid deep learning architecture for predicting diabetes and classifying severity levels. By combining temporal and geographical data aspects, the study presents a unique deep learning method that improves diabetes prediction by utilizing the convolutional gated recurrent unit (CGRU). It employs clustering methods in conjunction with a comprehensive framework that includes data preparation, model training, and evaluation to achieve accurate severity classification. This novel strategy goes beyond current practices to enhance the early diagnosis and detection of diabetes.

This research aims to employ the CGRU mechanism to extract spatial and temporal characteristics, enabling the detection of intricate patterns for diabetes diagnosis. Additionally, it seeks to identify the severity level of the condition, allowing for targeted and personalized treatment plans based on patients' specific needs (*Xie et al., 2024*). This

CGRU framework highlights its potential to improve early detection and diagnosis, ultimately enhancing healthcare outcomes. Moreover, the development of a web-based mobile-responsive application for diabetes prediction and severity level classification can improve the accessibility and real-world applicability of the proposed model. The following are the key contributions of the research:

- Development of a prediction model: This research develops a machine learning-based prediction model for instant diabetes detection, utilizing user-input features to improve accessibility and enable early diagnosis. This service allows individuals to easily assess their diabetes risk and take proactive steps toward better health through tailored risk assessments.

- Integration of the convolutional neural network-gated recurrent unit (CNN-GRU) model: To enhance diabetes prediction accuracy, a hybrid CGRU model is proposed, integrating convolutional neural networks with gated recurrent units. This model validates diabetes risk assessments by incorporating both temporal and spatial information from medical datasets, leading to more accurate predictions and improved patient outcomes.

- Clustering algorithm: To further refine the diabetes risk prediction model, a clustering algorithm is employed to address the complexities of diabetes-related data, enhancing the model's reliability and accuracy.

- Performance outcome: The proposed method achieves high accuracy rates, outperforming traditional machine learning techniques.

- The study starts with an introduction in the "Introduction" that highlights the value of proactive healthcare management and describes the relevance of using a mobile app to forecast diabetic illness. "Related Works" provides a comprehensive overview of the study by reviewing prior studies and literature on mobile health applications and diabetes prediction. The problem identified from the existing research is described in "Research Gap". The CGRU model is proposed in the "Materials and Methods" to describe the model architecture and the methods used to predict diabetes using a mobile app. The "Results and Discussion" presents the outcomes of the experiments conducted using the proposed technique. The study's major conclusions are finally outlined in the "Conclusion", along with their implications for diabetes prediction using mobile technologies and potential areas for future research.

## Related works

Recent research demonstrates improvements in the application of machine learning for predicting diabetes, including in areas such as logistic regression, random forests, and deep learning domains. Nevertheless, the classification of severity levels and integration of these models into operational, user-friendly systems remain largely understudied in the existing literature. The literature review of this study identifies gaps in current models, notably in handling spatial-temporal data for the severity prediction and improving access through more user-friendly and responsive online platforms.

*Maniruzzaman et al. (2020)* developed a machine learning (ML) system for diabetes individual prediction. The risk factors for diabetic sickness are identified by logistic regression (LR), which makes use of the *p*-value and odds ratio (OR). The research uses four different machine learning algorithms: "naïve Bayes (NB), Ada Boost (AB), random forest (RF), and decision tree (DT)." These algorithms were tested across 20 trials using three different partitioning methods. The area under the curve and accuracy are used to assess these classifiers' capabilities. The overall accuracy of the ML-based approach is 90.62%. The connection between LR-based choosing characteristics and RF-based categorization for the K10 procedure results in an accuracy of 94.25% ACC and an AUC of 0.95. As a result, the LR and RF-based classifications function more effectively now. The two of these will be extremely beneficial in the prediction of people with diabetes. The study's medical data categorization structure has a drawback in that it may be difficult to adapt for use with other medical data classification schemes, potentially restricting its future usefulness and efficacy in meeting the needs of doctors and patients.

*Ahmad et al. (2021)* compare the functions of fasting plasma glucose (FPG) and HbA1c as input characteristics in forecasting diabetes in patients. By reducing the health and financial costs associated with diabetes, the ability to predict a patient's condition based on a few key variables can enable quick, easy, and affordable diabetes screening. The research established acceptable results on the training set by utilizing five distinct artificial intelligence classifiers and feature removal through feature permutations and hierarchical clustering. This suggests that our information or characteristics are not limited to particular models. Research suggests that key determinants unique to the Saudi population can be identified through illness analysis utilizing particular characteristics and managing these factors may help reduce the disease. Research also offers some suggestions based on the findings of this study. The study's limitations include its restricted emphasis on Saudi Arabia and the possibility of using larger, more diverse datasets in future research to investigate deep learning and other black-box approaches.

*Battineni et al. (2019)* stated that the main cause of death and a prevalent chronic illness is diabetes. Patients with diabetes who receive an early diagnosis have the chance to properly control their condition by changing their lifestyle and eating habits. Numerous investigations have explored the prediction and diagnosis of this illness using ML methods. This study utilized the Pima Indian Diabetes Dataset (PIDD), which includes data from 768 female patients. Different data mining approaches were used to examine four different machine learning classifiers: "logistic regression (LR), RF, J48, and NB." The models were analyzed using different cross-validation configurations. Accuracy, F-score, precision, recall, and area under the curve (AUC) indicators of performance were computed individually for each model individually. The primary constraint of this research is that it only examined traditional machine learning classifiers. Despite potential advancements in existing diabetes prediction methods, research may encounter difficulties in studying unsupervised deep learning and machine learning methodologies due to the need for further exploration.

*Dey, Hossain & Rahman (2018)* have developed a method for forecasting and evaluating diabetes using several computer-based detection methods. The standard method for

diagnosing diabetic patients is more expensive and time-consuming. However, with advancements in machine learning, researchers are now able to provide a solution to this challenging issue. With an 82.35% prediction rate, the artificial neural network (ANN) shows a tremendous boost in accuracy, which motivates the creation of a web-based interactive application for diabetes prediction. One limitation of the work is that it builds a location-based dataset using real medical data and applies a deep learning model to predict diabetes. This might lead to problems with ethics and data privacy.

*Al Sadi & Balachandran (2023)* published an article on predicting Type 2 diabetes using an ANN and six machine learning classifiers. This article examines the detection of T2DM using artificial intelligence and the machine learning model on the Omani population using a custom dataset and six models: K-NN, SVM, NB, Decision tree, RF, LDA, and ANN, with MATLAB software (The MathWorks, Natick, MA, USA). The study analyzed data from the prediabetes register and the Al Shifa health system in South Al Batinah Province. The Random Forest and Decision Tree models achieved an accuracy of 98%. An important finding of the study was the observation that a gain in the number of features increases diagnosability, especially, in the case of missing values. Further work could involve conducting similar studies with different medical datasets and incorporating additional attributes to improve the results and system efficiency. However, shortcomings include the use of a particular database and a limited number of features.

*Kozinetz et al. (2024)* developed a model on nocturnal high and low-glucose prediction in adults with diabetes using both machine-learning and deep-learning models. This study used ML and deep learning (DL) models to predict the nocturnal glucose level in T1D patients who underwent MDI therapy. Specifically, the continuous glucose monitoring data were used in the training and testing of the models involving 380 subjects. Both MLP and CNN which are the parts of DL algorithms, as well as RF and GBTs, which are parts of ML algorithms, were tested. The models demonstrated high accuracy in predicting glucose levels within the target range (F1 metric). Within a 30-min prediction horizon, the proposed method achieved high accuracy, with F1 scores ranging from 93% to 97% and 96% to 98%, exceeding the target range. However, in the case of low glucose levels, the performance of these models was comparatively less accurate (F1: 80–86%) with MLP having the highest accuracy in these cases. The analysis also showed that both DL and ML models are useful for estimating the patients' nocturnal glucose levels in those with type 1 diabetes (T1D) using MDI ranges. More work must be done in the future to obtain larger groups with which to improve low glucose alerts and forecasts. Thus, assessing the utility of these predictive models in mobile applications for preventing FM and nocturnal hypoglycemia is still a significant issue to address.

*Alghamdi (2023)* conducted research on predicting the complications of diabetes using computational intelligence. The classification and prediction model uses data mining techniques to extract useful knowledge on diabetes data to assist in early detection and diabetes prediction. The XGBoost classifier, using a gradient boosting framework, has demonstrated a high accuracy rate in predicting diabetes from large datasets with many labeled features. However, the decision of the best algorithm predicting diabetes may depend on the nature of the collected data and the specific objectives of the study. In

addition to prediction, these techniques are useful for defining risk factors, tracking disease dynamics, and evaluating treatment outcomes. They provide essential information that helps the healthcare providers in their decision-making regarding disease processes. Further studies are required to implement other machine learning algorithms and data analysis techniques, as well as to extend the proposed XGboost classifier to improve accuracy and generalization with other benchmark datasets of clinical relevance (*Lugner et al., 2024*).

## Research gap

In recent years, diabetes management and prediction have seen substantial advancements through mobile applications and deep learning models. While extensive research has been conducted on diabetes risk assessment and control using ML and DL techniques, significant gaps remain in delivering accurate, generalized, and mobile-friendly solutions that integrate multi-dimensional variables essential for managing diabetes severity. The existing models face limitations in accurately predicting and grading diabetes severity due to the lack of integration across diverse data dimensions, such as lifestyle, genetic, and physiological factors, in a holistic and interpretable manner (*Ahmad et al., 2021*; *Maniruzzaman et al., 2020*). The current models often emphasize prediction without effectively addressing severity classification or continuous, on-the-go monitoring, leaving a gap in practical tools for diabetes self-management (*Kozinetz et al., 2024*; *Alghamdi, 2023*). Moreover, existing models often rely on conventional machine learning or single-layer deep learning approaches that fail to capture the intricate dependencies and temporal dynamics within these multi-dimensional data points, leading to reduced prediction accuracy and grading precision (*Alghamdi, 2023*). This research addresses these gaps by developing an integrated, mobile-accessible hybrid model for diabetes prediction and severity grading, leveraging multi-dimensional data in real-time, thereby advancing both the accessibility and functionality of diabetes management tools.

## MATERIALS AND METHODS

This section deals with the Proposed Hybrid CGRU model with K-means clustering for diabetes prediction and severity classification. Initially, it explains how the front-end and back-end interface works in the prediction process.

A subfield of artificial intelligence called machine learning enables computers to learn, make discoveries, and forecast results without the assistance of humans. Machine learning has been utilized in various sectors and is now being actively applied to develop mobile applications. Additionally, the evolving TensorFlow lite provides mobile apps developers with new and fascinating capabilities that are easy to use. Strong mobile machine-learning apps may use excellent business models and carry out intricate tasks. Retrofit, a REST API, was explored in this study (*Sarker, 2021*). The Python API communicates with the Java API, represented by Retrofit, to send and receive responses. Retrofit was chosen as the Java API in this study due to its user-friendly nature. The main advantage is that everyone can make API calls just as rapidly as they can make Java method calls. Its features, such as the ability to add custom headers and request types, upload files, and simulate responses, make

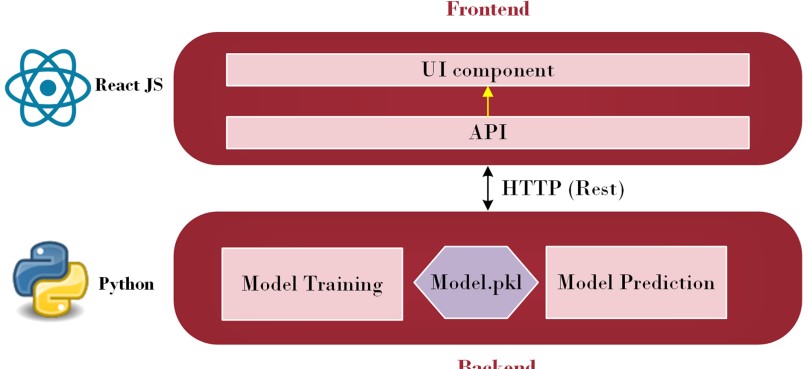

**Figure 1** **Front-end and Back-end Interface (Source: Python Logo (https://commons.wikimedia.org/ wiki/File:Python.svg) from *Wikiversity (2010)* and React JS logo (https://commons.wikimedia.org/ wiki/File:React_Logo_SVG.svg) from *Wikimedia Commons (2023)*).**

it easier to use web services while reducing boilerplate code in applications (*Miklosik & Evans, 2020*). Figure 1 illustrates the architecture employed to integrate a machine learning model into a smartphone.

As depicted in Fig. 1, this architecture consists of two primary parts: the front-end and the back-end. Through HTTP API Service, a request from the front end is sent to the back end. Once the data reaches the back-end, it is processed by the model depending on the input. The model then begins to predict and forwards the data over HTTP (REST) to the user interface (UI) component, from which the result is shown. Employing this process, the machine learning model integrates with the web-based mobile responsive application to finalize the forecast. React JS is used in the front-end development process to create an intuitive dashboard that allows users to observe their everyday activities. Meanwhile, diabetes prediction is handled by the Python back-end. Model prediction, a trained model file called "Model.pkl," and model training are some of its constituent parts. The trained model is used by the backend to estimate the chance of diabetes when users enter their daily data, ensuring smooth communication between the two levels. The phases of the proposed ML-based diabetes prediction and severity level classification are depicted in Fig. 2.

### Hybrid deep learning model for diabetes prediction

In the first step, the dataset preparation process is carried out based on the collected data. The given data is cleaned and processed, then split into training and testing datasets. Next, the pre-processed data is input into the Hybrid CGRU model, where features are extracted to predict diabetic conditions. Finally, the model's performance in predicting diabetes is evaluated using various performance metrics.

### *Data collection*

The BRFSS data collection, an open-source diabetes dataset, was originally compiled by the National Institute of Diabetes and Digestive and Kidney Diseases for use in the machine

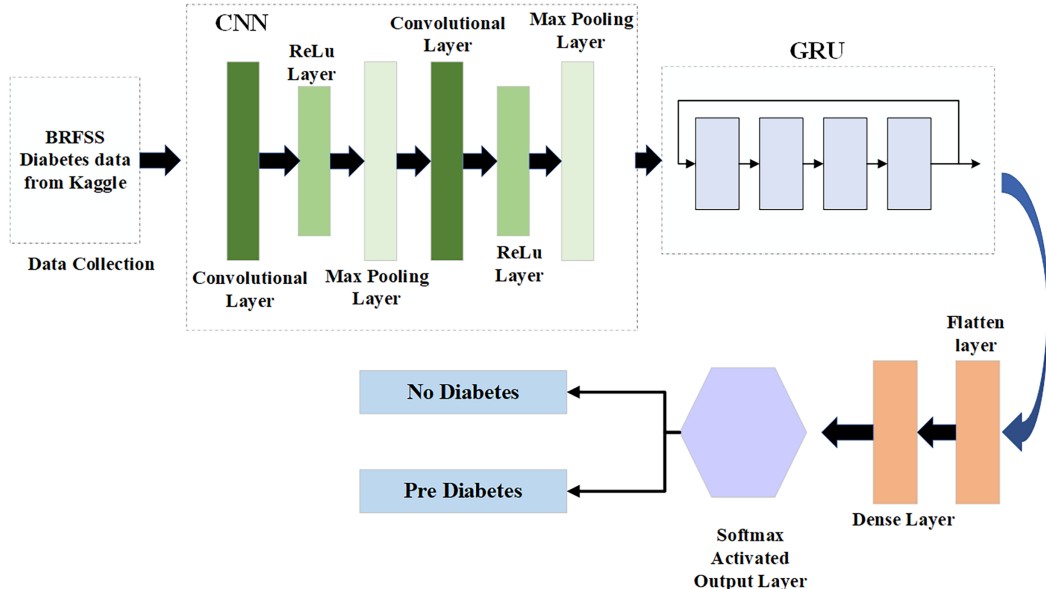

**Figure 2 Proposed model for diabetes prediction and severity level classification.**

learning categorization in this study (*kaggle, 2021*). Table 1 shows the description of the dataset used in this research.

## Data pre-processing

Data pre-processing is crucial for ensuring the model's accuracy and reliability. It involves cleaning, transforming, and preparing raw data for subsequent modeling and analysis.

### Missing data analysis

In this research, the dataset used is inherently balanced and does not contain any missing values, making it unnecessary to perform data imputation. By using the full dataset, researchers retained all the information necessary for analysis, model training, and diagnostics. This approach enhances the performance and robustness of predictions, avoiding the potential issues associated with imputed data, such as misleading information or additional biases. Therefore, the proposed model incorporates accurate input free from biases, and this enhances its ability to produce reliable and accurate predictions (*Phung, Kumar & Kim, 2019*). This ensures the information is maintained in its real form, retaining its integrity and structure for further analysis.

### Irrelevant feature removal

One of the critical steps in enhancing model performance involves redundancy and cleaning of the dataset through irrelevant feature removal. This research focused on reducing the number of attributes used in diabetes prediction to retain only the most impactful features. Recursive feature elimination (RFE) was thus applied, a feature selection technique that is robust in recursively eliminating lesser important features based

**Table 1 Description of datasets.**

| Key factors | Description |
| --- | --- |
| Age | The individual's age in years. |
| Glucose | The remaining amount of glucose in the blood 2 h after a meal, often called the "2-h postprandial blood sugar level" |
| Insulin | A person blood level insulin, measured in microunits per milliliter (µU/mL) |
| Blood pressure | Measurement, given in millimetres of mercury (mm Hg), of the pressure of blood on artery walls as it circulates throughout the body. |
| HighBP | A binary variable that shows if a person has high blood pressure |
| High Cholesterol | A binary value that indicates if a person has elevated cholesterol. |
| Smoker | A binary variable that indicates if someone smokes |
| Stroke | A binary variable that indicates whether or not a person has had a stroke |
| BMI | A body fat and health status measurement that is computed by dividing weight in kilograms by height in meters squared |

on a measure of importance determined by the impact on prediction outcome (*Bari & Karande, 2021*). Based on this, the seven attributes from the original dataset—'Fruits,' 'Veggies,' 'HvyAlcoholConsump,' 'NoDocbcCost,' 'DiffWalk,' 'Education,' and 'Income'— were deemed irrelevant to the analysis and were subsequently excluded. This helped filter out factors that, while concerning, were not closely linked to predicting the probability of diabetes, such as BMI, high blood pressure, and physical activity. This unique feature selection process using RFE resulted in a more enhanced model, highly suited for classifying the severity of diabetes.

### Data cleaning

Data cleaning is a crucial step to ensure the reliability and quality of the dataset for accurate diabetes severity prediction. In this research, duplicate records were identified and removed to prevent bias caused by over-representing certain observations, ensuring each record contributed equally to the model's learning process. These data-cleaning measures collectively ensured a high-quality dataset, enabling the deep learning model to achieve superior accuracy in predicting and classifying diabetes severity levels.

### Normalization

Normalization is applied to convert the numeric features of the dataset to a common scale, minimizing stability and convergence issues in the model. To help overcome possible biases, the data is normalized using the Z-score normalization method to bring the scale of each feature closer to zero mean and unit variance. Batch normalization was applied during model training to improve learning and further enhance the accuracy of diabetes severity classification (*Tan et al., 2023*). The normalization layer in the batch normalization approach is defined as follows in Eq. (1) and performs Z-normalization on the output of the preceding layer (*Yazdanian & Sharifian, 2021*). Using Eq. (1), the min–max scaling can be achieved.

$$A_{scaled} = \frac{A - A_{min}}{A_{max} - A_{min}} \tag{1}$$

A is the dataset's starting value, $A_{min}$ is the smallest value, and $A_{max}$ is its highest value. This technique might be helpful when the features are not evenly distributed or fall within a limited range.

**Data transformation**

Data transformation is one of the functional parts of the process of preparing the dataset for analysis. To improve feature extraction and reduce feature space, where features are somewhat related, aggregation techniques have been used. Generalization was aimed at reducing complexities by combining values of similar categories, while smoothing was also used to enhance dataset readability by avoiding distortions. These steps improved the general quality and organization of the data, providing a better chance of predicting the severity of diabetes (*Karim, Majumdar & Darabi, 2019*).

### *Deploying hybrid CGRU deep learning model for diabetes prediction*

Compared to LSTM architectural design, the GRU design requires fewer parameters to be configured and is more straightforward (*Chung et al., 2014*). Consequently, optimizing the CNN model using GRU makes sense, as CNN and GRU, while not identical, can effectively complement each other. This study develops a new CNN-GRU model. The input and output architecture of the CNN-GRU system proposed in this study is shown in Fig. 3.

**Convolutional layer**

The convolution layers gather the features from the input data, which is made up of many convolution kernels. Every cell of the convolution kernel is associated with a bias vector and a weight factor. Every convolutional layer neuron has connections to several neighboring neurons in the vicinity of the preceding layer. The dimensions of the area are determined by the size of the convolution kernel, commonly referred to in the literature as the "receptive field." The significance is comparable to the visual cortex's receptive field. The input characteristics are frequently scanned while the convolution kernel is operating. They are subsequently multiplied and averaged by the matrix components of the receptive field, and the variations are applied on top of each other (*Wu et al., 2023*).

$$R^{a+1}(x,y) = \left[R^a \otimes s^{l+1}\right](x,y) + a = \sum_{k_l}^{g=1} \sum_{g}^{h=1} \sum_{g}^{r=1} \left[R_l^k(w_0 x + a w_0 y + r)s_{l+1}^k(u,v)\right] + a \tag{2}$$

$$(x,y)\varepsilon\{0,1,\ldots,H_{l+1}\}, \ H_{l+1} = \frac{H_l + 2q - g}{w_0} + 1. \tag{3}$$

In Eq. (2), the summation part defines the equivalent to cracking a cross-correlation and defines the deviation. The input and output of the convolution layer of $l+1$ are denoted as $R^{a+1}$ and $R^a$, respectively, and are also known as a feature maps. It is assumed that the

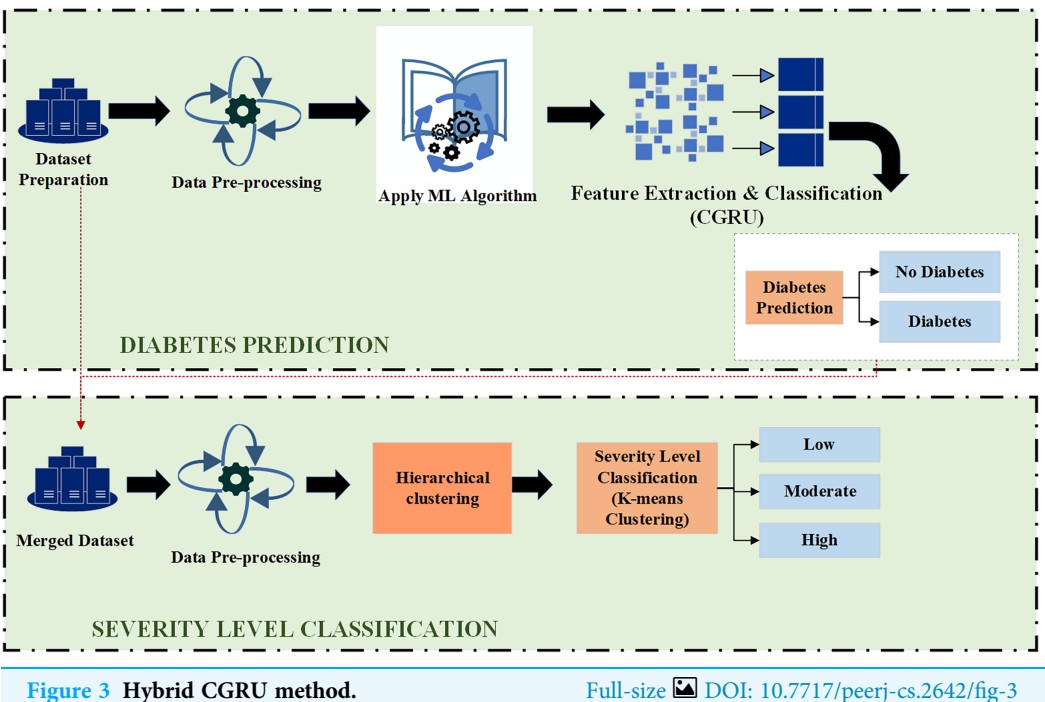

**Figure 3 Hybrid CGRU method.**

feature maps, in this case, have the same length as well as width since $R + 1$ is the size of $R^{a+1}$ in Eq. (3). The pixel of the feature map is defined as R(a,b) and the number of characteristic graph is mentioned as K. The parameter of the convolution layer of consistent to the side of the convolution step, convolution kernel and padding layer are defined as $w_0$, $g$ and $q$, respectively.

The settings of the convolution layer include its length and width. The output feature map size of the convolution layer, which is a CNN hyperparameter, depends on the kernel size. The kernel size is specified as a value smaller than the total dimension of the image input. Overall, the convolution layer is a key setting in the CNN. There are multiple possible sizes for the convolution kernel. A bigger convolution kernel maintains its lower size relative to the input picture while extracting more information from the input and enhancing model performance. The convolution phase sweeps the feature map twice and determines the distance between each convolution kernel point. Over each element of the feature map, the convolutional kernel is applied during the convolution phase with a size of 1, which can neglect $n − 1$ pixels when the number of steps exceeds 1. The convolutional layer has an activation function designed to highlight intricate features and their presentation. The equation is shown in Eq. (4):

$$M_{x,y}^{l} = f\left(R_{x,y,k}^{l}\right) \tag{4}$$

where '$x$' and '$y$' represent the position of the convolution layer at the position in the $l$th layer and $k$th filter and the activation function is denoted as "$f$".

One of the most common activation functions is called Relu. This often means that the curved function is represented using different types and slope functions. It is described as Eq. (5):

$$g(e) = max\ (0, e). \tag{5}$$

In the above Eq. (5), the $g(e)$ represents the output of ReLUs activation function for input is denoted as '$e$' which is the maximum of "0" and the input "$e$".

## Max-pooling layer

The pooling layer prevents overfitting in the output by down-sampling the input vector. The simulation's complexity of computation is decreased by the pooling layer. An evaluation of the pooling layer's output is given as follows:

$$a_i^k = down\ \left(a_i^{k-1}, s\right) \tag{6}$$

where $s$ is the pool size, $a_i^k$ is the characteristic vectors of the preceding layer, and down $(k-1)$ is the downsample as mentioned in Eq. (6). Two frequently used procedures for pooling are maximum pooling and average pooling.

## Gated recurrent unit

Using an embedding layer (explained below), the multi-hot encoded input $x$ is transferred into a low-dimensional embedding, as shown in Fig. 4. In the last stage, a fully connected layer is created by combining the patient's demographic data vector with fully interconnected layers that have a hyperbolic tangent activation and the hidden state at the final timestamp. On top of the patient's results, an additional fully connected layer (also known as the logistic regression layer) is implemented.

This layer computes the patient's risk score by using a single neuron with sigmoid activity. The following Eqs. (7) to (11):

$$Reset\ Gate\ r_g = \sigma\left(W^{(au)}\bar{c}_t + W^{(hv)}h_{t-1}\right). \tag{7}$$

In Eq. (7), $r_g$ represents the reset gate value with the sigmoid activation function "$\sigma$", the weight matric "$W^{(au)}$" for the input $\bar{c}_t$ in the reset gate. $\bar{c}_t$ define the current time step. The weight matrix of the previous hidden state "$h_{t-1}$" is defined by $W^{(hv)}$.

$$Update\ Gate\ z_g = \sigma\left(W^{(au)}\bar{c}_t + W^{(hv)}h_{t-1}\right) \tag{8}$$

$$Process\ Input\ \tilde{h}_g = \tanh\left(W^{(i\tilde{h})}\bar{c}_t + W^{(h\tilde{h})}h_{t-1}\right) \tag{9}$$

$$Hidden\ State\ h_g = (1 - v_t) * h_{t-1} + z_t * \tilde{h}_t \tag{10}$$

$$Output\ State\ O_s = h_t. \tag{11}$$

In Eqs. (7) to (11), $z_g, \tilde{h}_g, h_g, O_s$ denotes the update gate, process input, hidden state, and output state, respectively. The tanh function represents the tangent activation function,
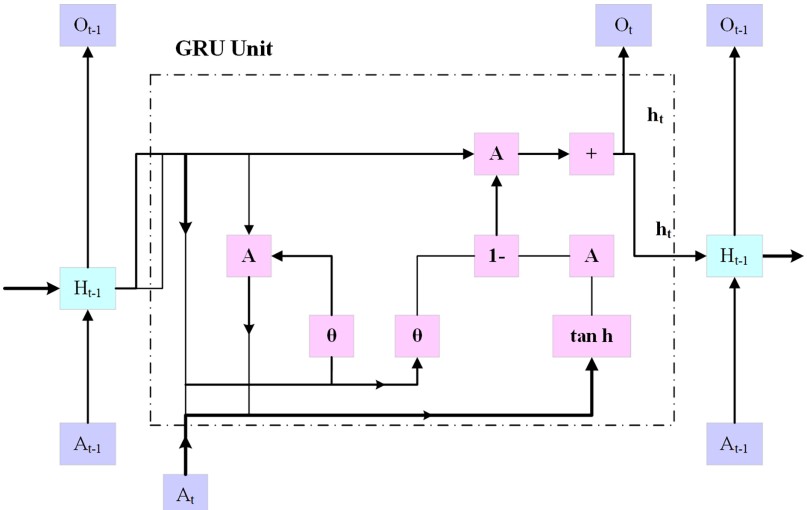

**Figure 4 GRU architecture.**

$W^{(i\tilde{h})}$ denotes the weight matrix for the processed input, and $W^{(h\tilde{h})}$ denotes the weight matrix for the previous hidden state. The CGRU approach presented in this work extracts certain important characteristics while maintaining the initial feature layout using a convolutional layer. Furthermore, to avoid the issue of the framework being over-fitted, the weak value feature was excluded by using the max pooling layer, and the intense feature value was selected from the significant feature. To eliminate eigenvalues less than zero, a rectified linear unit is applied in the convolutional layer, while a max pooling layer is incorporated to enhance the training efficiency. Subsequently, to speed up the simulation's computation while enhancing its accuracy, the eigenvalues are transmitted *via* the GRU update and reset gates.

**Flatten layer**

One of the key components of the CNN is the fully connected layer, which integrates the spatial characteristic refined by the pooling layer that follows the convolutional layer. Using the dense connection, the fully connected layer enables the fusion of learned information across the spatial dimension, with each neuron connected to every neuron in the preceding layer. This integration phase needs to capture complex relationships between the input so that the network can identify complicated patterns ranging from lower-level features such as edges and textures to higher-level abstraction. The integration is mathematically represented by matrix multiplication, followed by bias addition and the application of the activation function. As a consequence, the network can provide accurate predictions for various machine-learning tasks. Before proceeding to the fully connected layers, the feature maps that the pooling and convolution layers have recovered must be flattened into a one-dimensional vector. The multi-dimensional feature maps are reshaped into a single long vector using this flattening technique. Every element in this vector represents a distinct quality that the convolutional layers were able to identify. By connecting every neuron in one layer to the other layer, the fully connected layer allows the

network to learn the complex representation. In a fully connected layer, every neuron has a unique function in identifying different facets of the input data and identifying certain patterns, or combinations of patterns, that are essential for outcome prediction. In many tasks involving classification, the output of the last fully connected layer is processed through a softmax activation function to produce probability ratings for each class.

**Softmax layer**

The softmax layer is a crucial component commonly utilized in neural network categorization jobs. Its primary function is to create an ordered distribution using the probabilities that result from the layers that precede it. This transformation ensures that the probabilities of all possible outcomes sum to one, thereby accurately assigning a likelihood to each class, such as "diabetes" and "no diabetes." The softmax layer normalizes the probabilities allowing the algorithm to select the class with the highest probability as the model's output and produce an accurate prediction.

The overall working process of the CGRU model in predicting diabetes is given below in Algorithm 1.

## Clustering algorithm for severity level classification

The clustering method is used to classify individuals into three severity classifications relative to diabetes: high, moderate, and low. Patient data, including demographics, medical histories, and laboratory test results, make up the dataset utilized for this research. The data is pre-processed to make sure it is in a format that is appropriate for clustering. Using the hierarchical clustering technique, patients are grouped into clusters based on their similarity. The process begins by assigning each patient to an individual cluster, which are then iteratively merged based on similarity until a single cluster remains. This technique identifies the optimal number of clusters and uses a dendrogram to visualize the hierarchical structure of the clustering results. The average severity level of the patients in each cluster, which is ascertained by the average value of the severity metric computed utilizing the patient data, is then employed to classify the clusters as high, moderate, or low severity categories.

### Hierarchical clustering

Hierarchical clustering is the most efficient technique for grouping data points into clusters based on their similarities. It detects patterns in severity levels and organizes them according to shared attributes or similarities. Different types of patients exhibit various abnormal behaviors, which may differ from one another. To classify the user with the same data characteristics, the research utilizes the hierarchical clustering algorithm. The fundamental stage in anomaly identification is to cluster the original diabetic data using the hierarchical clustering technique. To identify data outliers using the standard information model, researchers typically need to train a sufficient number of labeled data samples in the diabetes severity prediction. The important process of hierarchical clustering is to compute the distance between two kinds of data points by merging the algorithm and combining the two closest data points. This helps to calculate the distance

**Algorithm 1 CGRU for Predicting the diabetes.**

*Input: Dataset $D = \{(X_i, y_i)| \ i = 1, 2, \ldots, n\}$, // $X_i$ is feature vector, $y_i \in \{0, 1\}$ is the label representing Diabetes (1) or No Diabetes (0)*

*Output: Predicting the condition $y_{pred} \in \{0, 1\}$ for each test instance*

*Begin*

*For each instance $(X_i, y_i)$ in D*

    *Clean the data by handling missing values and outliers*

    *Normalize or standardize features in $X_i$ to ensure consistent scaling*

    *Perform feature selection by identifying relevant features and dropping irrelevant ones*

*end for*

*Split the dataset D into a training set $D_{train} = \{(X_{train}, y_{train})\}$ and a testing set $D_{test} = \{(X_{test}, y_{test})\}$ using a specified split ratio (80–20)*

*Train the data in the CGRU model*

*Initialize the CGRU model parameters*

*Define convolutional layer parameters, such as filter count and kernel size, for spatial feature extraction*

*Define GRU layer parameters to capture temporal patterns*

*For each epoch e*

    *For each batch $(X_{batch}, y_{batch}) \in D_{train}$*

        *Extract spatial features from $X_{batch}$ using the convolutional layers: $F_{spatial} = ConvLayer(X_{batch})$*

        *Pass $F_{spatial}$ through GRU layers to capture temporal features: $F_{temporal} = GRULayer(F_{spatial})$*

        *Predict $y_{pred}$ based on $F_{temporal}$*

        *Calculate the loss $L = Loss(y_{pred}, y_{batch})$ using the loss function*

        *Update the model parameters to minimize L using an optimizer*

    *End for*

*End for*

*Test the data in the CGRU model*

*For each instance $(X_{test}, y_{test}) \in D_{test}$*

    *Extract spatial features from $X_{test}$ using the convolutional layers: $F_{spatial} = ConvLayer(X_{test})$*

    *Extract temporal features from $F_{spatial}$ through GRU layers $F_{temporal} = GRULayer(F_{spatial})$*

    *Predict $y_{pred} \in \{0, 1\}$ based on $F_{temporal}$*

    *Compute loss and Update model parameters using backpropagation*

*End for*

*Evaluate the trained model on the test set (Accuracy, Precision, Recall, F1-score)*

*Print the test loss and test accuracy*

*End*

among each kind of data point and also determine the similarity among them. At first, each class is defined based on the sum of distortion degree. If the "$n$" samples were divided into '$k$' classes, with '$k$' being less than '$n$' and >2, then $Z_k$ represents the class $k$ ($k = 1, 2, \ldots, n$). The degree of distortion for this class is calculated using Eq. (12), where $\mu_k$ denotes the center of gravity:

$$\sum_{u \varepsilon Z_k} |a_u - \mu_k|^2. \tag{12}$$

The degrees of distortion of all classes are calculated using Eq. (13).

$$J = \sum_{k=1}^{n} \sum_{u \varepsilon Z_k} |a_u - \mu_k|^2. \tag{13}$$

$J$, often known as the aggregation coefficient, shows a progressive decline in its "$a_u$" value, with increasing cluster count. The ideal number of clusters is ascertained when the grouping coefficient begins to converge. Figure 5 shows the hierarchical clustering dendrogram where the data points are grouped based on similarity. The x-axis represents the individual data point and the y-axis represents the distance or dissimilarity between clusters. To differentiate the clusters, the branches are color-coded as blue, green, and red. The distance between clusters is determined by merging the heights of the two clusters. The workflow of the hierarchical clustering process is mentioned in Algorithm 2.

### K-means clustering algorithm

Using a Python-based combinatorial k-means clustering analysis tool, which requires the selection of a small number of descriptors from a larger set, the separation of instances was determined through k-means clustering. Utilizing the Python Notebook platform, the combinatorial k-means clustering program was created utilizing SciPy, Matplotlib, and Itertools. The flow chart of the clustering process is shown in Fig. 6. The patient data for each underlying illness was retrieved from spreadsheet files using the Pandas module and visualized with the Matplotlib tool. K-means clustering was performed using the hierarchy of the cluster submodule from SciPy. Every possible combination of any descriptor was examined using the Itertools package. The user-selected Excel file contained all the data extracted by the application. Subsequently, the data were divided by each attribute's standard deviation to normalize them. To perform the k-means clustering, several linking techniques are available, such as centroid, average, and Ward's.

The hierarchical clustering algorithm begins by initializing the process and taking the measured features of data points as input. It computes a distance matrix to show the pairwise distances between all points using metrics such as Euclidean or Manhattan distance. Initially, each data point is considered its cluster. The algorithm then checks if there is only one cluster. If not, it merges the two closest clusters, updates the distance matrix, and repeats the process until only a cluster remains. This iterative process continues, combining the most similar data points or clusters and updating distances, until all points are consolidated into a single cluster.

For all computations in this paper, the centroid approach was chosen. The program combined every conceivable combination of three descriptors from each of the spreadsheet's properties that were marked as input data, using k-means clustering by default. Combinations including two descriptors are an additional option. This study used artificial clustering to arrange the three variables into every feasible combination. For every

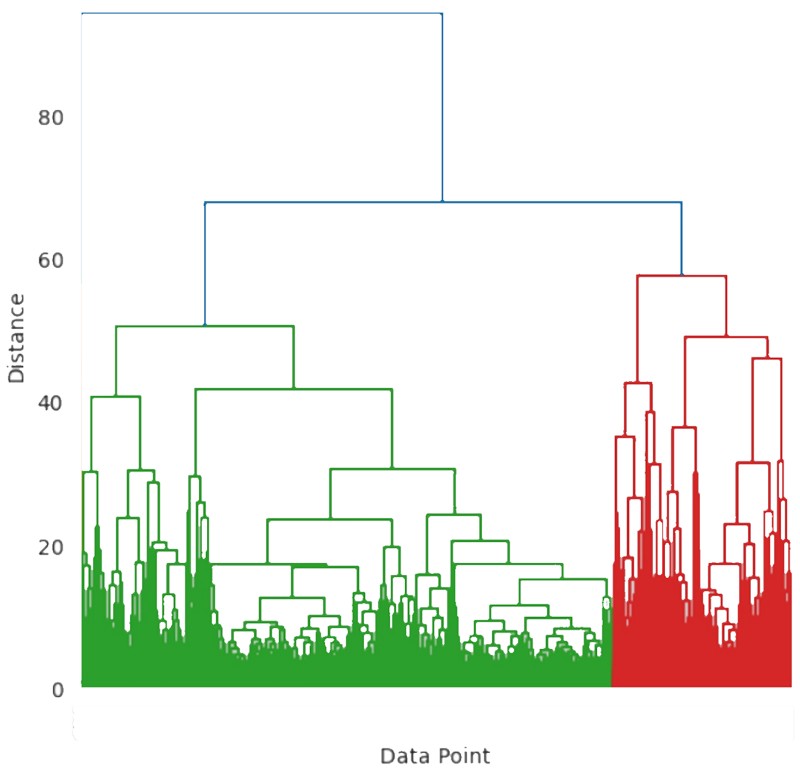

**Hierarchical Clustering Dendrogram**

**Figure 5 Hierarchical clustering dendrogram.**

k-mean clustering, a parameter known as the global variance was calculated to rank every feasible clustering.

$$varGlobal = \sum \sum var_{i,k}. \tag{14}$$

In Eq. (14), $var_{i,k}$ is the variance of the descriptor "$i$" for the "$k$" cluster and it can be also stated that each cluster has different descriptors. Some restrictions were also implemented to choose a legitimate clustering. The stochastic c clustering process may result in systems with a non-equilibrium number of cluster members, leading to clusters with one or two items that exhibit low variance within the associated group. It is specified that a legitimate clustering is a situation where at least three components for any of the clusters are involved to prevent this problem. Three different descriptor combinations were used by the program to achieve clustering. The right number of instances with data available for the three descriptors under investigation were chosen for each combination. In this manner, most data were always utilized for each clustering.

## RESULTS AND DISCUSSION

This section includes a description of the system and the results of the suggested diabetes prediction model. The collected dataset is implemented in the Python software in a Windows 10 Operating System. First, the efficacy of the CGRU method was assessed.

---

**Algorithm 2** Hierarchical clustering for grouping the severity class.

*For each instance $X_{diabetic} \in y_{pred}$*

    *Select relevant severity features such as blood sugar levels, HbA1c levels, and other indicators of severity*

*Apply K-means Clustering for Initial Classification*

    *Initialize K-means clustering with $K = 3$ (representing Low, Medium, and High severity levels)*

*Run K-means on the severity feature set to create initial clusters, clusters $= K - means(X_{diabetic}, K = 3)$*

    *Assign cluster labels to each instance based on proximity to cluster centroids*

        *Low - cluster with the lowest centroid value*

        *Medium - cluster with intermediate centroid value*

        *High - cluster with the highest centroid value*

*Perform hierarchical clustering on the initial clusters for further refinement*

    *Use an appropriate linkage criterion and distance metric*

    *Distance function $\{z_1, z_2\}$*

    *for $u = 1$ to n*

    *$z_u = \{a_u\}$*

    *end for*

    *$Z = \{z_1,...z_n\}$*

    *$U = n+1$*

    *While Z.size >1 do*

        *Find two closest cluster $Z_{min1}$, $Z_{min2}$ with minimum distance:*

        *–$(Z_{min1}, Z_{min2}) = min\ dist\ (z_u, z_v)$ for all $z_u, z_v$ in Z*

        *–eliminate $Z_{min1}$ and $Z_{min2}$ from Z*

        *–add the merged cluster $\{Z_{min1}, Z_{min2}\}$ to C*

        *–Increment $U = U+1$*

    *end while*

    *Check the dendrogram to verify natural groupings and refine boundaries between Low, Medium, and High-severity clusters*

*For each instance $X_{diabetic}$*

    *Assign final severity labels to each cluster outcome*

        *Low - cluster with minimal severity indicators*

        *Medium - cluster with intermediate severity indicators*

        *High - cluster with highest severity indicators*

    *output the predicted severity label: Low, Medium, or High*

    *end for*

*end for*

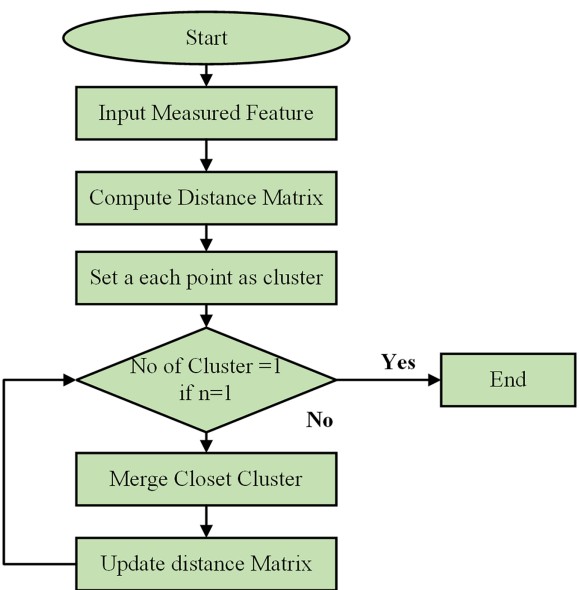

**Figure 6  Flowchart of clustering process.**     

Subsequently, the developed Android mobile application and website structure are presented.

## Experimental results for diabetes prediction and severity level classification

Figure 7 shows the input data of the binary classification of diabetes in which the different parameters that cause diabetes were used to evaluate the performance. Figure 8 illustrates the outcome of the missing value analysis and it shows that there are no/zero missing values present.

Figure 9 illustrates the result of the irrelevant feature removal, highlighting the characteristics of various health indicators, such as blood pressure, cholesterol level, BMI, and smoking status, *etc*. Based on that, the seven attributes from the original dataset, namely, 'Fruits,' 'Veggies,' 'HvyAlcoholConsump,' 'NoDocbcCost,' 'DiffWalk,' 'Education,' and 'Income' were considered as attributes that do not add any value to the analysis and were subsequently dropped from the analysis. The individual health data is presented in each row representing the absence of the specific condition.

Figure 10 shows the BMI *vs*. PhysHlth scatter plot, where the data points represent individual BMI and physical health status, providing details on the potential correlation between higher BMI, poor physical health, and increased diabetes risk.

The distribution of high cholesterol by age and sex is presented in Fig. 11. The prevalence of high cholesterol among the 'Sex 0.0' and 'Sex 1.0' is compared with the age group from "<10" to "<80" within each group. It can be obviously observed that there is a gradual increase in each case, with a sudden rise occurring in the age group '60–69'.

Figure 12 shows the comparison of mental health score distributions for individuals who have had a stroke. The person with a stroke is labeled as "1.0" and the non-stroke is

| | Diabetes_binary | HighBP | HighChol | CholCheck | BMI | Smoker | Stroke | HeartDiseaseorAttack | PhysActivity | Fruits | ... | AnyHealthcare | NoDocbcCost | GenHlth |
|---|---|---|---|---|---|---|---|---|---|---|---|---|---|---|
| 0 | 0.0 | 1.0 | 0.0 | 1.0 | 26.0 | 0.0 | 0.0 | 0.0 | 1.0 | 0.0 | ... | 1.0 | 0.0 | 3.0 |
| 1 | 0.0 | 1.0 | 1.0 | 1.0 | 26.0 | 1.0 | 1.0 | 0.0 | 0.0 | 1.0 | ... | 1.0 | 0.0 | 3.0 |
| 2 | 0.0 | 0.0 | 0.0 | 1.0 | 26.0 | 0.0 | 0.0 | 0.0 | 1.0 | 1.0 | ... | 1.0 | 0.0 | 1.0 |
| 3 | 0.0 | 1.0 | 1.0 | 1.0 | 28.0 | 1.0 | 0.0 | 0.0 | 1.0 | 1.0 | ... | 1.0 | 0.0 | 3.0 |
| 4 | 0.0 | 0.0 | 0.0 | 1.0 | 29.0 | 1.0 | 0.0 | 0.0 | 1.0 | 1.0 | ... | 1.0 | 0.0 | 2.0 |

5 rows × 22 columns

**Figure 7 Input data.**

```
Missing values in the datase
Diabetes_binary          0
HighBP                   0
HighChol                 0
CholCheck                0
BMI                      0
Smoker                   0
Stroke                   0
HeartDiseaseorAttack     0
PhysActivity             0
Fruits                   0
Veggies                  0
HvyAlcoholConsump        0
AnyHealthcare            0
NoDocbcCost              0
GenHlth                  0
MentHlth                 0
PhysHlth                 0
DiffWalk                 0
Sex                      0
Age                      0
Education                0
Income                   0
dtype: int64
```

**Figure 8 Missing value analysis outcome.**

labeled as "0.0". The spread and density of mental health scores in each group are shown in the Y-axis, ranging from −5 to 35. This helps identify patterns or differences in mental health conditions associated with stroke occurrence.

Figure 13 illustrates the correlation matrix heatmap, showing the strength and direction of the relationships between variables. Positive correlation is represented in green, negative correlation in red, and no correlation is indicated in yellow. This color-coding helps in interpreting correlation values ranging from −1 to 1.

| | Diabetes_binary | HighBP | HighChol | CholCheck | BMI | Smoker | Stroke | HeartDiseaseorAttack | PhysActivity | AnyHealthcare | GenHlth | MentHlth | PhysHlth |
|---|---|---|---|---|---|---|---|---|---|---|---|---|---|
| 0 | 0.0 | 1.0 | 0.0 | 1.0 | 26.0 | 0.0 | 0.0 | 0.0 | 1.0 | 1.0 | 3.0 | 5.0 | 30.0 |
| 1 | 0.0 | 1.0 | 1.0 | 1.0 | 26.0 | 1.0 | 1.0 | 0.0 | 0.0 | 1.0 | 3.0 | 0.0 | 0.0 |
| 2 | 0.0 | 0.0 | 0.0 | 1.0 | 26.0 | 0.0 | 0.0 | 0.0 | 1.0 | 1.0 | 1.0 | 0.0 | 10.0 |
| 3 | 0.0 | 1.0 | 1.0 | 1.0 | 28.0 | 1.0 | 0.0 | 0.0 | 1.0 | 1.0 | 3.0 | 0.0 | 3.0 |
| 4 | 0.0 | 0.0 | 0.0 | 1.0 | 29.0 | 1.0 | 0.0 | 0.0 | 1.0 | 1.0 | 2.0 | 0.0 | 0.0 |
| ... | ... | ... | ... | ... | ... | ... | ... | ... | ... | ... | ... | ... | ... |
| 70687 | 1.0 | 0.0 | 1.0 | 1.0 | 37.0 | 0.0 | 0.0 | 0.0 | 0.0 | 1.0 | 4.0 | 0.0 | 0.0 |
| 70688 | 1.0 | 0.0 | 1.0 | 1.0 | 29.0 | 1.0 | 0.0 | 1.0 | 0.0 | 1.0 | 2.0 | 0.0 | 0.0 |
| 70689 | 1.0 | 1.0 | 1.0 | 1.0 | 25.0 | 0.0 | 0.0 | 1.0 | 0.0 | 1.0 | 5.0 | 15.0 | 0.0 |
| 70690 | 1.0 | 1.0 | 1.0 | 1.0 | 18.0 | 0.0 | 0.0 | 0.0 | 0.0 | 1.0 | 4.0 | 0.0 | 0.0 |
| 70691 | 1.0 | 1.0 | 1.0 | 1.0 | 25.0 | 0.0 | 0.0 | 1.0 | 1.0 | 1.0 | 2.0 | 0.0 | 0.0 |

58033 rows × 15 columns

**Figure 9  After removing irrelevant features.**     

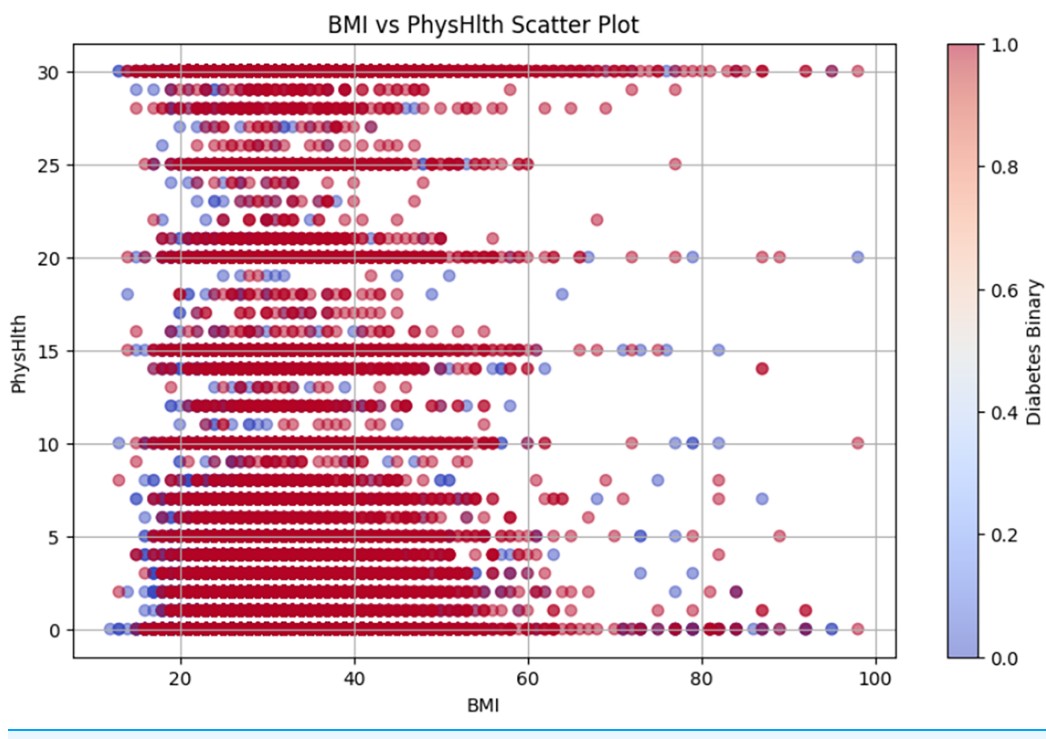

**Figure 10  BMI *vs* PhysHlth scatter plot.**     

The Layer explanation of the proposed neural network architecture is presented in Fig. 14. It includes the layers such as "conv1d_3", "max_pooling1d_3", "flatten_3", "dense_7", "dense_8" and "dense_9" and each layer represents the output shape. The convolution layer includes 128 parameters contributing to the total sum of 16,705 trainable parameters, which are updated during training.

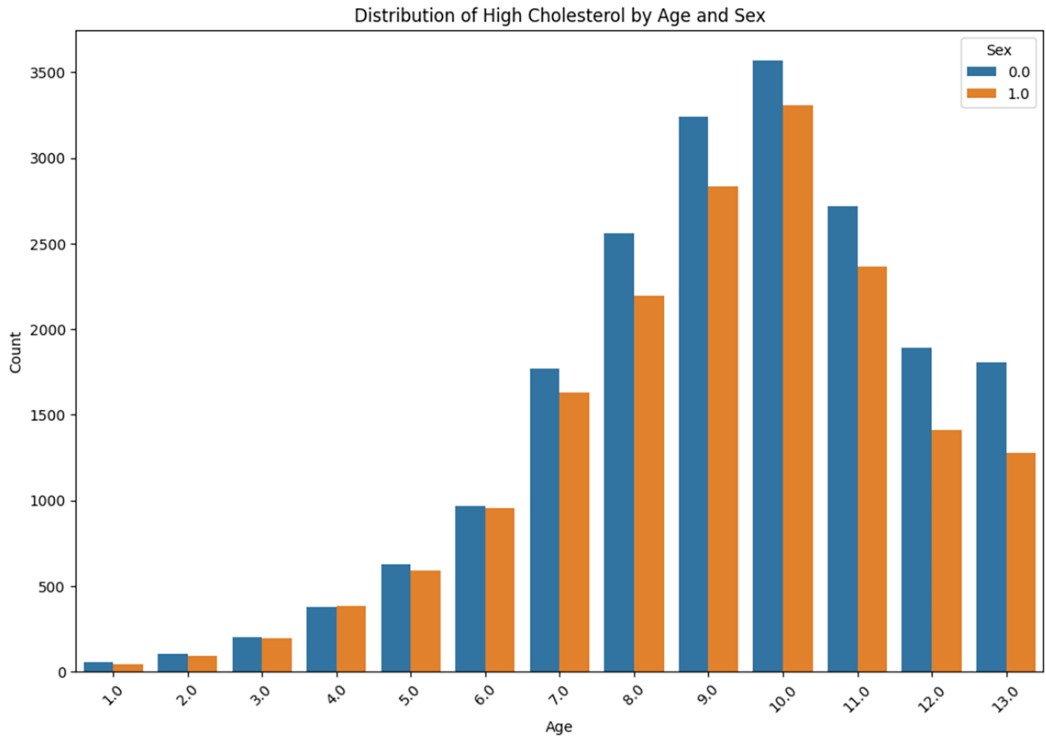

Figure 11 **High cholesterol by age and sex.**     

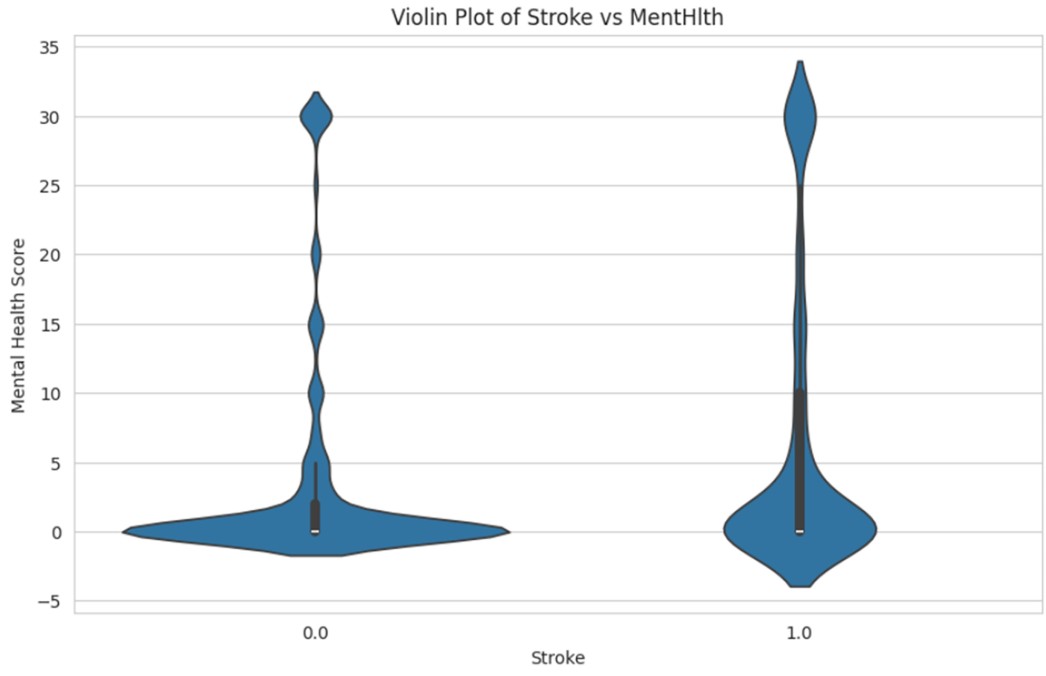

Figure 12 **Violin plot of stroke *vs.* mental health.**     

Figure 15 depicts the graphical representation of "training and validation loss" and "training and validation accuracy" for 20 epochs. For training and validation loss for each epoch, the loss is gradually reduced to 0.002%, and for the training and validation accuracy, it is gradually raised to 99%.

Figure 16 presents the ROC curve for the proposed model, which is used to evaluate the efficiency of the binary classifier. It plots the true positive rate against the false positive rate at various thresholds. The curve indicates efficient performance between the classes, with a perfect classification achieved, and an AUC of 0.99 (≈1).

The diabetes-predicted data taken for Severity analysis is shown in Fig. 17 with key health indications such as "Diabetes", "Hunger", "BMI", and "Cholesterol", among others. The numerical values were defined based on their health metrics.

Figure 18 shows the severity level group for the taken data as moderate, high, and low levels. Among the three sets of severity levels taken from the dataset, the moderate level has the highest rate, followed by highest level lies in second place, and the lowest level in third.

The performance of a machine learning algorithm in the form of a confusion matrix is presented in Figs. 19A and 19B. From Fig. 19A, it can be observed that there are 10,604 true positives, 0 false positives, 1 false negative, and 10,603 true negatives. Based on this, it can be concluded that the high number of correct predictions and the negligible number of errors indicate high prediction accuracy. Similarly, from Fig. 19B, it is evident that there is a high count along the diagonal, indicating a high level of accurate predictions. In conclusion, the classification results demonstrate high accuracy, minimal errors, and improved efficiency.

## Performance evaluation

The evaluation metrics are the quantitative measures used to determine the ML model's performance and efficacy. These metrics provide information about how well a model is doing and help compare different models or algorithms. The metrics used for evaluation and its formula are provided below.

Accuracy: It is the percentage of correctly classified instances out of the total occurrences, reflecting the model's overall prediction performance. Equation (16) may be used to calculate the accuracy.

$$Accuracy = \frac{T_p + T_N}{T_p + T_N + F_p + F_N} \tag{16}$$

Recall: A classification model's recall measures its ability to correctly identify all pertinent examples. It is calculated as the ratio of true positive predictions to the total number of genuine positive occurrences. It is also known as a true positive rate or sensitivity at times. The recall might be determined using Eq. (17).

$$Recall = \frac{T_p}{T_p + F_N} \tag{17}$$

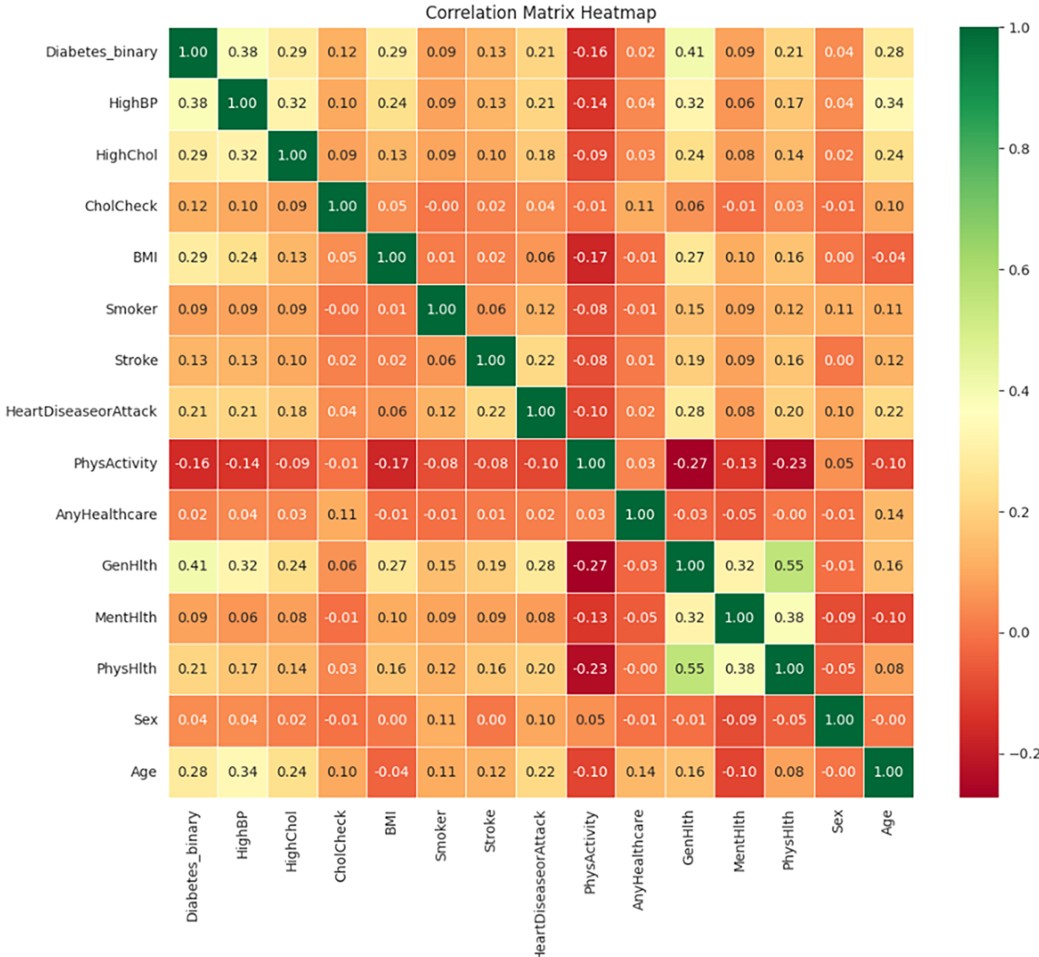

**Figure 13  Correlation matrix heatmap.**  

```
Layer (type)                    Output Shape            Param #
================================================================
conv1d_1 (Conv1D)               (None, 13, 32)          128

max_pooling1d_1 (MaxPoolin      (None, 6, 32)           0
g1D)

flatten_1 (Flatten)             (None, 192)             0

dense_2 (Dense)                 (None, 64)              12352

dense_3 (Dense)                 (None, 64)              4160

dense_4 (Dense)                 (None, 1)               65

================================================================
Total params: 16705 (65.25 KB)
Trainable params: 16705 (65.25 KB)
Non-trainable params: 0 (0.00 Byte)
```

**Figure 14  Neural network architecture.**  

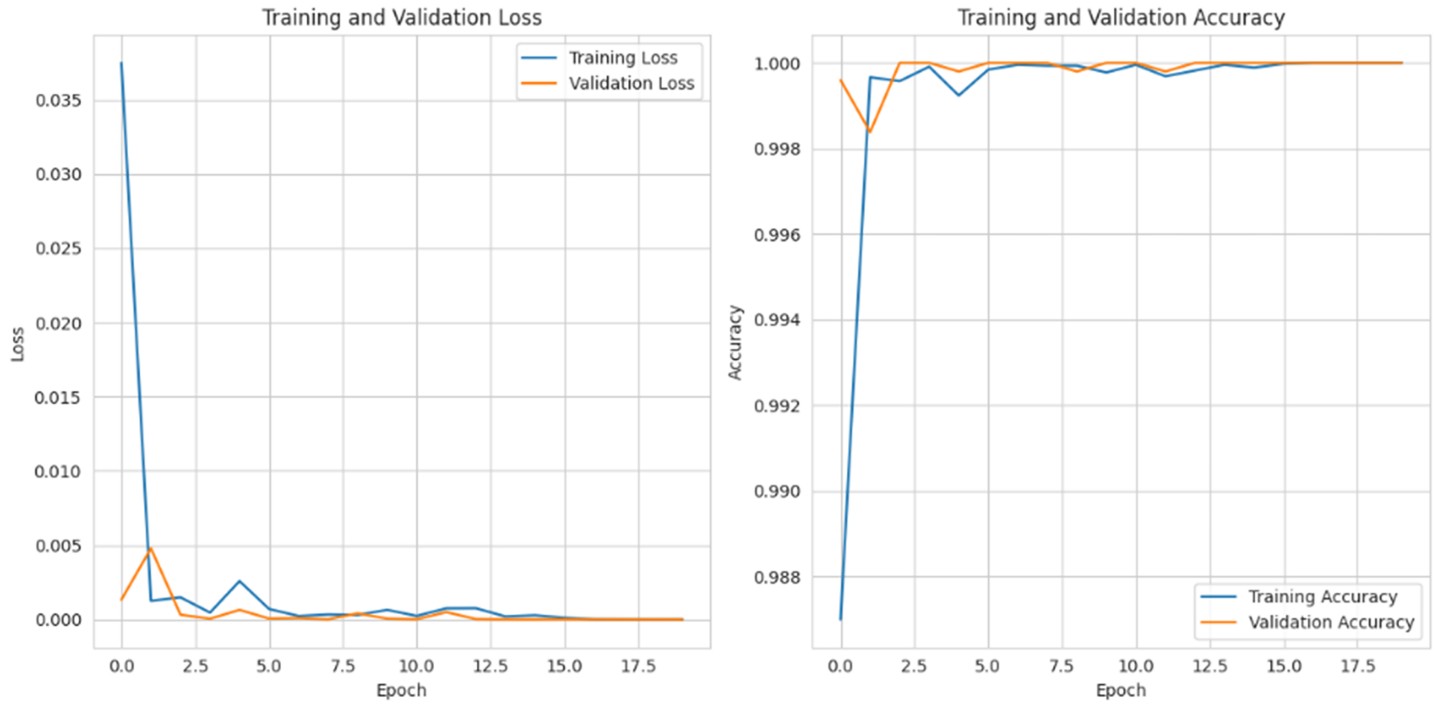

**Figure 15 Training and validation loss and accuracy.**

**Figure 16 ROC curve.**

| | Diabetes_binary | HighBP | HighChol | CholCheck | BMI | Smoker | Stroke | HeartDiseaseorAttack | PhysActivity | AnyHealthcare | GenHlth | MentHlth | PhysHlth |
|---|---|---|---|---|---|---|---|---|---|---|---|---|---|
| 35346 | 1.0 | 1.0 | 1.0 | 1.0 | 30.0 | 1.0 | 0.0 | 1.0 | 0.0 | 1.0 | 5.0 | 30.0 | 30.0 |
| 35347 | 1.0 | 0.0 | 0.0 | 1.0 | 25.0 | 1.0 | 0.0 | 0.0 | 1.0 | 1.0 | 3.0 | 0.0 | 0.0 |
| 35348 | 1.0 | 1.0 | 1.0 | 1.0 | 28.0 | 0.0 | 0.0 | 0.0 | 0.0 | 1.0 | 4.0 | 0.0 | 0.0 |
| 35349 | 1.0 | 0.0 | 0.0 | 1.0 | 23.0 | 1.0 | 0.0 | 0.0 | 1.0 | 1.0 | 2.0 | 0.0 | 0.0 |
| 35350 | 1.0 | 1.0 | 0.0 | 1.0 | 27.0 | 0.0 | 0.0 | 0.0 | 1.0 | 1.0 | 1.0 | 0.0 | 0.0 |

**Figure 17 Severity data pre-processed result.**

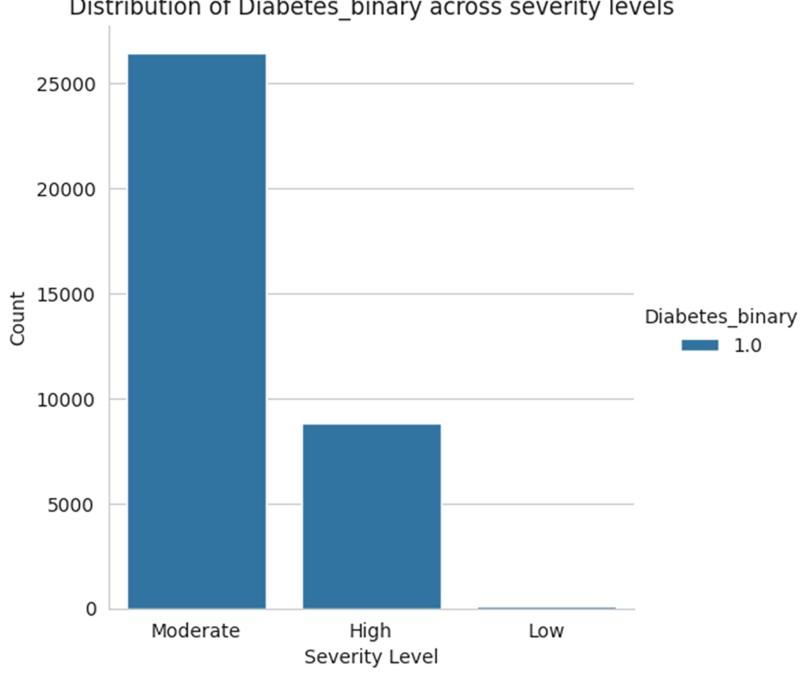

**Figure 18 Severity levels of diabetes prediction.**

Precision: The effectiveness of a classification model's predicted outcomes is measured by its precision. It is calculated as the ratio of the overall number of accurate positive predictions to the entire number of effective predictions made by the model. Mathematically, it is expressed by Eq. (18).

$$Precision = \frac{T_p}{T_p + F_p}.$$
(18)

F1-score: The F1-score is determined by taking the harmonic mean of accuracy and recall. It is a statistical measure that offers a balance between the two parameters. As it accounts for both false positives and false negatives, it is particularly useful in situations where there is an unequal distribution of classes. F1-score is calculated using Eq. (19).

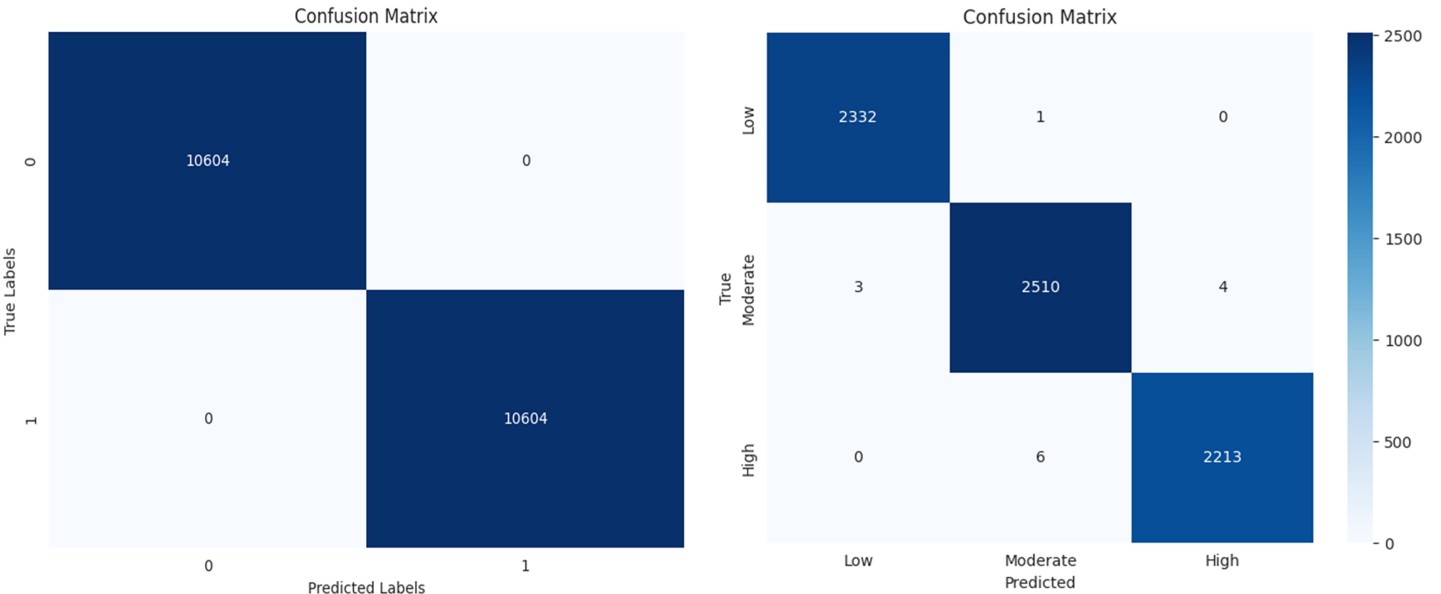

**Figure 19 Confusion matrix for diabetes prediction and severity classification.**

$$F1 - score = 2 \times \frac{Precision \times Recall}{Precision + Recall}. \tag{19}$$

In Eqs. (16), (17), and (18), $T_p$ and $T_N$ represent the true positives and true negatives, while $F_N$ and $F_p$ refer to the false negatives and positives, respectively.

Table 2 shows the performance evaluation of the proposed CGRU model with the Clustering algorithm in predicting diabetes and classifying the severity stage. From Table 2, it can be concluded that the proposed model provides enhanced accuracy, precision, recall, and an F1-score of 99.99%, correspondingly.

Table 3 and Fig. 20 present the comparison of severity prediction of Existing research (*Zhao et al., 2024*; *Dutta et al., 2022*) and the proposed model. It can be concluded that the proposed model achieves high accuracy of 99.9%, while the ensemble ML classifier shows a lower accuracy of 73.4%. Figure 21 shows the interface of the proposed web-based mobile responsive application in diabetes prediction. This improvement shows the better ability of the model to extract the spatial and temporal characteristics from the dataset improving the forecast accuracy. They further note that the complex patterns make it difficult for the model to learn, suggesting that the CGRU framework is more reliable for accurate classification of the level of severity of diabetes.

## DISCUSSION

The most significant finding of this study is the remarkable performance of the proposed Convolutional Gated Recurrent Unit (CGRU) model for diabetes prediction, and severity level classification, achieving an outstanding accuracy, precision, recall, and F1-score of 99.9%. This exceptional performance highlights the model's robustness and potential to

**Table 2 Performance metrics of proposed model.**

| Metrics | Efficiency |
|---|---|
| Accuracy | 99.9% |
| Precision | 99.9% |
| Recall | 99.9% |
| F1-Score | 99.9% |

**Table 3 Comparison of proposed method with different methods.**

| References | Method | Accuracy |
|---|---|---|
| *Zhao et al. (2024)* | Attention based CNN | 94.12% |
| *Dutta et al. (2022)* | Ensemble ML Classifier | 73.4% |
| Proposed model | CGRU with K-means clustering | 99.9% |

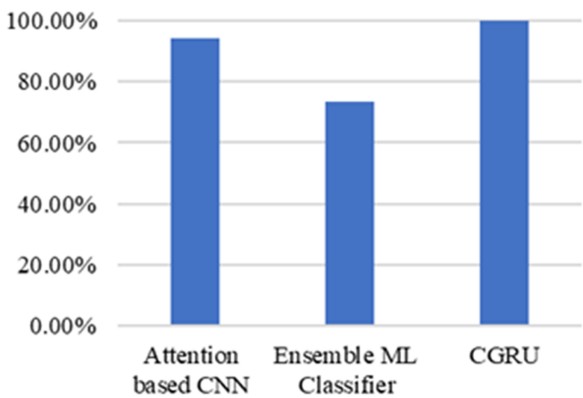

**Figure 20 Performance comparison of proposed method with existing models.**

enhance diabetes care by enabling precise and timely predictions. The integration of this model into a web-based mobile-responsive application further enhances its usability and accessibility, making it a practical tool for real-time deployment and user-friendly interaction. This directly addresses the gap in previous studies, which often lacked the integration of mobile-responsive features and accurate severity classification, thereby improving the practical utility and applicability of the model in everyday healthcare settings.

When comparing the results to existing methods, such as the Attention-based CNN (94.12% accuracy) (*Zhao et al., 2024*) and the Ensemble ML Classifier (73.4% accuracy) (*Dutta et al., 2022*), the superiority of the CGRU model is evident. Previous studies faced challenges related to dataset dimensions and population specificity, which limited their generalizability and effectiveness. The proposed model effectively overcomes these

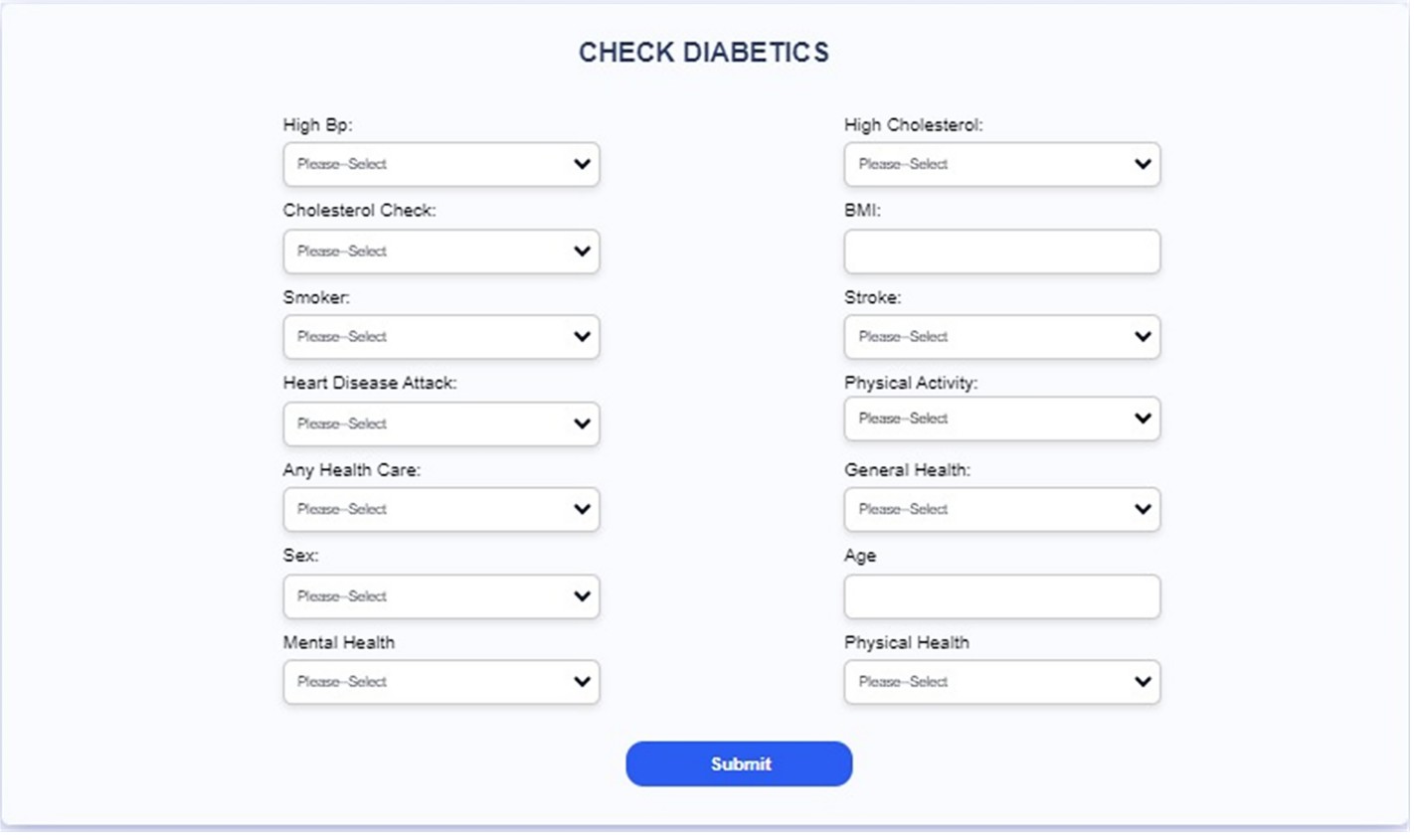

Figure 21 **Interface of the proposed web-based mobile responsive application.**

limitations by capturing both temporal and spatial information from medical datasets through the integration of CNN and GRU. Additionally, the use of a clustering algorithm enhances the model's reliability and accuracy, addressing the complexities inherent in diabetes-related data by classifying the severity levels of diabetes, which is not present in the existing applications. This comprehensive approach not only improves prediction accuracy, but also offers significant advancements in patient outcomes by providing tailored diabetes risk assessments and management strategies.

Despite the promising results, this study has certain limitations. One primary concern is the need for further validation across diverse datasets to ensure the model's generalizability and robustness. Additionally, the model's performance in real-world scenarios with varying data quality and noise levels remains to be thoroughly tested. Unexpectedly, the CGRU model achieved higher precision and recall rates than initially anticipated, indicating its effectiveness in minimizing false positives and negatives. Further studies should be aimed at replicating the proposed model in other samples to eliminate the possibility of overfitting the model. Furthermore, to produce accurate results, it is crucial to evaluate the efficiency of the adopted model by performing trials with real-world data of diversified quality and with different levels of noise included. It will strengthen the validity of the model if similar performance can be demonstrated in other clinical settings and with

less homogeneous data. However, future research should explore the robustness of the model in different clinical environments and with more heterogeneous data to fully establish its reliability and applicability.

## CONCLUSION

This work develops an innovative architecture for deep learning for the early diagnosis and estimation of diabetes mellitus: the CGRU. By integrating gated recurrent units with convolutional layers, the CGRU model effectively extracts both temporal and spatial properties from input data, closing significant gaps in existing methods. A useful technique for assembling comparable data points according to the similarities is the Hierarchical Clustering method. Each data point is initially treated as a separate cluster. The method then iteratively identifies and merges the two closest clusters, adding the combined cluster to the resulting set. This process is repeated until only one cluster remains. By carefully pre-processing the BRFSS dataset, which included imputation of missing data, elimination of superfluous features, and normalization, the researchers were able to obtain high-quality input for model training. The CGRU model surpassed two popular machine learning algorithms, SVM and CNN-LSTM, in terms of prediction accuracy, with a rate of 99.9%. The proposed model performs well due to the balanced dataset, which is implemented in clinical practice, ensuring its reproducibility. The proposed model delivers improved prediction results on balanced datasets and can be incorporated into a web-based mobile application for the early diagnosis of diabetes mellitus and more effective intervention. Potential directions for future work involve extended investigation of the CGRU framework in the context of other chronic diseases, enlargement of a dataset used for the analysis to make it more representative, improvement of model configurations, optimization of user interfaces, and long-term research concerning the effects on the health status of patients and overall healthcare costs. This development is expected to enhance the ability to diagnose, preview, and manage diabetes, offering valuable resources for healthcare workers and individuals affected by the disease. Future studies could explore the applicability of the CGRU framework to other chronic diseases, expand the variety of the dataset to improve the model's generalizability, and refine the model's design to enhance accuracy. Improvements to the web-based application's user interface, along with longitudinal studies to evaluate the long-term impact of early diabetes prediction on patient health outcomes and healthcare costs, would also be valuable. By integrating this model into a web-based mobile-responsive application, we have created a practical tool for early diagnosis and timely intervention, thereby enhancing patient outcomes and improving healthcare delivery.

### Funding

This project was funded by the Deanship of Scientific Research (DSR) at King Abdulaziz University, Jeddah, under grant no. (GPIP: 1046-611-2024) and the Universiti Teknologi, Malaysia for the financial sponsorship of the research through the UTM Encouragement

Research Grant (UTMER), grant reference number/no: PY/2022/03968; cost center: Q.J130000.3828.31J51. The funders had no role in study design, data collection and analysis, decision to publish, or preparation of the manuscript.

### Grant Disclosures

The following grant information was disclosed by the authors:
King Abdulaziz University, Jeddah: GPIP: 1046-611-2024.
Universiti Teknologi, Malaysia: PY/2022/03968 and cost center: Q.J130000.3828.31J51.

### Competing Interests

The authors declare that they have no competing interests.

### Author Contributions

- Alhuseen Omar Alsayed conceived and designed the experiments, performed the experiments, analyzed the data, performed the computation work, prepared figures and/or tables, authored or reviewed drafts of the article, and approved the final draft.
- Nor Azman Ismail conceived and designed the experiments, authored or reviewed drafts of the article, and approved the final draft.
- Layla Hasan performed the computation work, prepared figures and/or tables, authored or reviewed drafts of the article, and approved the final draft.
- Muhammad Binsawad conceived and designed the experiments, performed the experiments, analyzed the data, performed the computation work, authored or reviewed drafts of the article, and approved the final draft.
- Farhat Embarak performed the experiments, analyzed the data, authored or reviewed drafts of the article, and approved the final draft.

### Data Availability

The raw data is available in the Supplemental File and Kaggle: https://www.kaggle.com/datasets/alexteboul/diabetes-health-indicators-dataset?select=diabetes_012_health_indicators_BRFSS2015.csv.

The BRFSS data collection, an open-source diabetes data set, was initially gathered by the "National Institute of Diabetes and Digestive and Kidney Diseases" for use in the machine learning categorization in this study.

### Supplemental Information

Supplemental information for this article can be found online at http://dx.doi.org/10.7717/peerj-cs.2642#supplemental-information.

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
