# Peer review of "Leveraging a hybrid convolutional gated recursive diabetes prediction and severity grading model through a mobile app"

_PeerJ Computer Science, doi:10.7717/peerj-cs.2642_

## Round 0.1 · original submission · Major Revisions

Thank you for submitting your manuscript, After careful evaluation by our review team, we appreciate the effort and depth of research that has gone into your work. However, we regret to inform you that the manuscript requires major revisions before it can be considered for publication.

The reviewers have identified several key areas that need significant improvement, including clarity of the methodology, robustness of the results, depth of the literature review, and the professional language of the manuscript. We believe addressing these concerns will strengthen the quality and impact of your research.

Please carefully review the attached comments from the reviewers and make the necessary revisions. We look forward to receiving your revised manuscript.

Reviewer 1 ·

Basic reporting

1. Title of the paper presents the initial impression so it should be appropriate, concise, precise, interesting and unique. I suggest to rewrite the title, it is too long, not interesting, it should be written in concise and precise terms.
2. The authors have good background knowledge of the work undertaken. The introduction section is quite adequate in contents however the introduction and importance of AI and machine learning is overemphasized instead of identifying the gaps in scientific knowledge and furnish strong justification for the current research. I suggest to brief the part of introduction section from Line #61 to Line#96. Highlight the gaps and issue in previous research give the justification of proposed research using CGRU. Also discuss how/what benefits and values the approach will bring to the area of research?

Experimental design

3. Give overview of the progress that has been done so far in the area of research and what are you specifically looking through literature review conducted in this study rather than directly discussing the prior researches.
4. Literature review justifies the scope of work undertaken in this study. However, the limitations are not identified after reviewing the literature. Instead of writing problem statement as a separate section, add a paragraph at the last of Literature Review section describing the failures and limitations of previous research studies.
5. The methodology demonstrated is understandable but need minor improvements for example at the outset you should discuss the major sections of the proposed methodology and then discuss each section with pertinent details referring to graphical representation of methodology.

Validity of the findings

6. Results interpreted in the light of proposed research objectives and existing literature. Includes alternative
explanations and instructional tables and graphs. The interpretations based on analysis are quite convincing.

7. The discussion section adequately summarized but the justification is not given of achieving such a high result, identical against each performance measure. Limitations are discussed but future research pathways to tackle such limitations are not highlighted.

Reviewer 2 ·

Basic reporting

The paper presents an insightful approach to addressing the critical issue of diabetes prediction and severity level classification through the implementation of a Convolutional Gated Recurrent Unit (CGRU). The authors developed a methodology that integrates advanced machine learning techniques with a user-friendly, web-based, mobile-responsive application, making it a valuable tool for both healthcare professionals and patients. However, there some corrections or flaws I observed in the paper that is stated below:

Authors should reframe the abstract to clearly outline the problem, the methodology (including the use of Convolutional Gated Recurrent Unit), the results, and the implications of the study.

Authors should highlight more on the novelty of the paper and its contributions to knowledge in the introduction section. The introduction section should be strengthened by discussing the significance of diabetes prediction and the challenges involved. Authors should consult more recent studies (23-2024) in this regards.

I suggest the authors carefully read and improve the readability of this paper alongside the language. The grammar ambiguous. There are series of grammatical errors which requires extensive reediting by a native English Reader. Also, since the study has been conducted, author should make use of past tense in the writings.

Authors should ensure that all figures and tables are properly labeled and referenced in the text.
The conclusion section should be reframed to succinctly summarize the findings and their implications, particularly the utility of your web-based application for healthcare professionals and patients. Authors should conclude with a statement on the potential impact of their work on diabetes prediction and management.

Experimental design

The methodology is sound and fair

Validity of the findings

The topic is novel and the findings are valid.

Additional comments

As above

·

Basic reporting

1. The abstract provides a general overview of the paper, touching on the significance of diabetes prediction and the use of Convolutional Gated Recurrent Unit (CGRU) models. It successfully introduces the problem and the aim of the study. However, author should enhance the structure of the abstract section. It should include a brief statement on the practical implications of the findings, such as how the web-based application can benefit healthcare professionals and patients. This will provide a more compelling summary of the paper's value.

Experimental design

The results section presents the performance metrics of the CGRU model clearly, with appropriate use of tables and figures. However, authors should consider discussing any limitations or areas where the model did not perform as well. This will help in understanding the model's limitations and areas for future improvement.

Validity of the findings

The paper offers a valuable contribution by integrating CGRU with a user-friendly application for diabetes prediction, a novel approach that addresses a critical healthcare need. However, it should be more explicit in highlighting what makes this work unique compared to existing research. For instance, how does the CGRU model improve upon other models in terms of accuracy or robustness in diabetes severity classification?
The literature review covers a range of studies related to diabetes prediction and machine learning applications in healthcare. However, it fails more clearly to identify the gaps in current literature that the study aims to address. Authors should highlight the recent studies in the related field which can lead to more clarity of research coverage.

Additional comments

The methodology section is well-structured and provides a clear explanation of diabetes prediction model. It will be efficient if more details were provided on the data preprocessing steps, including how missing data, or imbalanced classes were handled. This will give readers a better understanding of how the data was prepared for modeling.

---

## Round 0.2 · Minor Revisions

Dear author

Thank you for your re-submission and for incorporating the comments. Although the reviewers are now happy with the revised updated version, we request you to please edit your manuscript and make the following necessary changes before we proceed.

1. Your rebuttal letter does not seem to be a point-by-point response to the comments raised by the experts. Therefore, please provide a detailed response letter.

2. Unfortunately, there are lots of language and grammatical errors in the manuscript roughly more than 800 issues. Therefore, it is strongly suggested to carefully proofread from a professional service. eg few examples
Abstract line 1:
illnesses connected to the high rate of could be illnesses connected to a high rate of
Abstract line 2:
there is need for early detection of diabetes could be there is a need for early detection of diabetes

Abstract Line 4:
which may lead to the difficulties in creating dependable and accurate could be
which may lead to difficulties in creating a dependable.

Research gap and the problem statement should be concise
every segment of the section
A. Hybrid Deep Learning model for Diabetes Prediction eg data collection, preprocessing should be explained in detail in the context of your study, not a generic definition.
Algorithm 1 should be written in a professional way (scientific writing)

Reviewer 2 ·

Basic reporting

The quality of paper has improved

Experimental design

Design is okay

Validity of the findings

Results have improved

Additional comments

Paper can be accepted

·

Basic reporting

The authors have addressed the essential elements of Basic Reporting by providing an abstract that effectively outlines a general overview of the study, highlighting the significance of diabetes prediction and the application of Convolutional Gated Recurrent Unit (CGRU) models. Additionally, they have responded to the need for structural enhancement in the abstract by including a brief statement on the practical implications of their findings. By explaining how the web-based application could support healthcare professionals and benefit patients, the authors offer a more compelling and complete summary of the study’s value and real-world relevance.
The authors have successfully addressed the basic reporting.

Experimental design

After reviewing the updated manuscript, it is clear that the authors have successfully addressed the need to discuss limitations or areas where the model's performance was less optimal. By including these details, they provide a more nuanced understanding of the CGRU model's capabilities and constraints. This addition not only enhances the transparency of the results but also offers valuable insights into potential areas for refinement, guiding future research and model improvement efforts.

After reviewing the updated manuscript, it is evident that the authors have successfully enhanced the clarity and rigor of their methodology by providing more detailed information on the data preprocessing steps. They have included explanations on how they addressed issues such as missing data and handled imbalanced classes, ensuring the robustness of their dataset before model training. This level of detail strengthens the study’s transparency and replicability, allowing readers to better understand the preprocessing techniques used to improve data quality and model performance.

Validity of the findings

After reviewing the updated manuscript, it is evident that the authors have successfully enhanced the validity of their findings by addressing key areas for improvement.
The authors have clearly identified the gaps in the current literature that their study aims to address, demonstrating a strong understanding of the field's existing limitations. They effectively set the stage for their research objectives and underscore the necessity of their approach. This clarity enhances the manuscript’s relevance, as it situates their work within the broader context of ongoing studies and highlights the unique contributions their findings offer to advance knowledge in the field.

Additional comments

The updated manuscript demonstrates significant improvements in both clarity and methodological transparency. It is reflecting the authors' careful consideration of previous feedback. The inclusion of additional details on data preprocessing and model limitations provides a more comprehensive understanding of the study's framework and results. Overall, the manuscript presents a valuable contribution to the field and is substantially strengthened by the recent updates.

---

## Round 0.3 · Minor Revisions

Dear authors

Unfortunately the revised paper still has lots of issues in language quality and other comments.

Therefore, it is required to get it professionally edited by a proofing service and also attach the language certificate with the re submission.

Thank you

---

## Round 0.4 · accepted · Accept

Thank you for making the language changes and providing the language editing certificate as well. Your paper is being recommended for publication